# Assessment of Long Lived Isotopes in Alkali-Silica Resistant Concrete Designed for Nuclear Installations

**DOI:** 10.3390/ma14164595

**Published:** 2021-08-16

**Authors:** Daria Jóźwiak-Niedźwiedzka, Katalin Gméling, Aneta Antolik, Kinga Dziedzic, Michał A. Glinicki

**Affiliations:** 1Institute of Fundamental Technological Research, Polish Academy of Sciences, Pawińskiego 5b, 02-106 Warsaw, Poland; aantolik@ippt.pan.pl (A.A.); kdzie@ippt.pan.pl (K.D.); mglinic@ippt.pan.pl (M.A.G.); 2Nuclear Analysis and Radiography Department, Centre for Energy Research, 29–33 Konkoly Thege Miklós Street, H-1121 Budapest, Hungary; gmeling.katalin@ek-cer.hu

**Keywords:** alkali-silica reaction, concrete durability, low-level radioactive waste, neutron activation analysis, radiation shielding concrete, trace elements

## Abstract

The design of concrete for radiation shielding structures is principally based on the selection of materials of adequate elemental composition and mix proportioning to achieve the long-term durability in nuclear environment. Concrete elements may become radioactive through exposure to neutron radiation from the nuclear reactor. A selection of constituent materials of greatly reduced content of long-lived residual radioisotopes would reduce the volume of low-level waste during plant decommissioning. The objective of this investigation is an assessment of trace elements with a large activation cross section in concrete constituents and simultaneous evaluation of susceptibility of concrete to detrimental alkali-silica reaction. Two isotopes ^60^Co and ^152^Eu were chosen as the dominant long-lived residual radioisotopes and evaluated using neutron activation analysis. The influence of selected mineral aggregates on the expansion due to alkali-silica reaction was tested. The content of ^60^Co and ^152^Eu activated by neutron radiation in fine and coarse aggregates, as well as in four types of Portland cement, is presented and discussed in respect to the chemical composition and rock origin. Conflicting results were obtained for quartzite coarse aggregate and siliceous river sand that, despite a low content, ^60^Co and ^152^Eu exhibited a high susceptibility to alkali-silica reaction in Portland cement concrete. The obtained results facilitate a multicriteria selection of constituents for radiation-shielding concrete.

## 1. Introduction

Biological shielding structures in nuclear power plants are exposed to ionizing radiation during their service life. After such exposure, certain parts of shielding structures may become a source of decay radiation, thus producing the low-level radioactive waste that has to be properly disposed of during the decommissioning of nuclear power plants. Assessment of the activation level of concrete components is required for the proper selection of methods of waste disposal. If the activation criteria were taken into account during construction at the stage of concrete mix design, the amount of radioactive waste could probably be significantly reduced. However, the selection of concrete constituent materials of highly reduced content of long-lived residual radioisotopes will not diminish the essential load-bearing and shielding function of concrete, nor impair its durability.

According to recommendations of the International Atomic Energy Agency [1], the material is classified as radioactive waste due to its clearance level (CL). Almost in all cases more than one radionuclide is involved, so the mixture of radionuclides below CL is expressed as $\overset{n}{\underset{i=1}{\sum }}\frac{c_{i}}{c_{li}}<1.0$, where $c_{i}$ is the mass specific activity of radionuclide *i* (Bq/g), $c_{li}$ is the clearance level of radionuclide *i* (Bq/g), *n* is the number of radionuclides in the mixture [2]. When the $\overset{n}{\underset{i=1}{\sum }}\frac{c_{i}}{c_{li}}$ of waste is less than 1, the waste can be treated as non-radioactive waste [3].

The dominant long-lived residual radioisotopes induced in ordinary concrete at the time of decommissioning, which occupy 99–100% of the total residual radioactivity in terms of CL value, are ^60^Co, ^152^Eu and ^154^Eu [4,5]. The representative single value of the clearance level and half-life are, respectively, CL^60^Co = 0.3 Bq/g, T_1/2_ = 5.275 years, CL^152^Eu = 0.3 Bq/g, T_1/2_ = 13.54 years, and CL^154^Eu = 0.3 Bq/g, T_1/2_ = 8.59 years. The profile of isotopes in spent nuclear fuel has been estimated by Gauld and Ryman [6]. They stated that the actual contribution of each isotope significantly varied based on the source of fuel and the time since discharge, but isotopes ^60^Co and ^154^Eu accounted for almost 50% of all notable isotopes [7]. Alpha, beta, and neutron particles are substantially absorbed by the casing, whereas a substantial fraction of the gamma rays passes through and impact the interior faces of the concrete in dry casks.

The recognized causes of concrete deterioration in nuclear environment were discussed by [8,9,10]. They mainly concerned the degradation of the reinforced concrete structure, which manifested itself in cracking, spalling or the delamination of cover concrete [8]. Besides thermal cracking [9,10], shrinkage [9], creep [9], carbonation and chemical aggression [9], the potential deterioration due to alkali-silica reaction (ASR) should be taken into account [11]. ASR is a chemical reaction between the reactive silica in fine or coarse aggregate and the alkalis (K^+^ and Na^+^) and hydroxyl (OH^−^) ions present in the concrete pore solution. If, during the concrete design process, the use of low-alkali cement or supplementary cementitious materials [12,13,14,15]—which mitigate the reaction as well as non-reactive aggregate—was not taken into account, the reaction is moving forward. This type of internal concrete damage progresses all the time, and there is no way to interrupt this reaction. ASR causes concrete expansion and cracking, significantly favoring other deterioration processes, especially regarding concrete tightness. This type of concrete degradation is likely the leading cause of pavement [16] and dam concrete deterioration. Alkali silica reaction has been identified as a concrete degradation mechanism for nuclear power plants in Canada [17]. ASR was discovered at Seabrook nuclear power plant in the USA, 25 years after plant construction [18]. ASR has also occurred in the structural concrete that forms the ring-beam, wall, dome and buttresses of the Gentilly 1 containment structure in France [19]. 

This phenomenon may have serious implications for the structural integrity and serviceability of nuclear power plants aging [20] and there are no currently available assessment criteria or guidelines to assess the consequences. 

In the context of intended nuclear power development in Poland an experimental program was undertaken to improve tools for concrete mix optimization by incorporating new criterion of the content of long-lived residual radioisotopes in concrete constituents. The objective of this investigation is to assess the content of residual radioisotopes of Cobalt and Europium (^60^Co and ^152^Eu) in principal concrete constituents and evaluate the susceptibility of concrete to detrimental alkali-silica reaction. The range of investigation covered several normal weight and heavyweight rock aggregates and Portland cements as concrete constituents. Using the neutron activation analysis and ASR expansion tests an enhanced evaluation of mineral aggregates is expected to support optimal mix design.

## 2. Materials and Methods

### 2.1. Materials 

Seven fine aggregates and nine coarse aggregates were selected for analysis. One crushed limestone sand S1 (ρ = 2.66 g/cm^3^), and six siliceous sands, S2–S7, were tested as fine aggregate. Sands S2 and S3 were characterized as natural river sand (ρ = 2.74 and 2.86 g/cm^3^, respectively), and S4–S7 as natural fossil sands (ρ = 2.63, 2.64, 2.63 and 2.65 g/cm^3^, respectively). Aggregate representing three main types of rock (igneous, sedimentary and metamorphic) and commonly used in concrete technology was selected as coarse aggregate; additionally, a heavy aggregate—baryte—was taken into consideration. The maximum aggregate size of coarse aggregate was 22.4 mm. Description of the aggregate origin and their density are presented in Table 1.

Aggregate samples were taken from the original landfill prior to further processing. The quantity of the taken samples was 25 kg for each fraction of the aggregate. For further tests, the entire sample was crushed and then sorted in order to ensure proper homogeneity. Smaller quantities of aggregate were selected by the quartering method.

Four Portland cements CEM I 42.5R and 52.5R were selected for analysis. Three of them were ordinary Portland cements differed in alkali content (C1, C2 and C3) and the fourth one, white cement with low content of iron was chosen (C4). The ordinary cements were from Polish Górażdże (C1), Małogoszcz (C2) and Norwegian Norcem (C3) cement plants, while white cement C4 was from Danish cement plant Aalborg. The chemical composition of cements is presented in Table 2; the notation follows the European standard PN-EN 197-1 [22]. The loss of ignition was determined according to PN-EN 196-2 [23]. 

Cement CEM I 52.5R with 0.88% Na_2_O_eq_ (C2) was used for the estimation of the aggregate potential alkali-reactivity tested according to accelerated mortar-bar test method and long-term concrete prism test. The cement fineness, determined using the PN-EN 196-6 [24] method, amounted to 525 m^2^/kg. The le Chatelier method (PN-EN 196-3 [25]) was used to estimate the cement soundness. The increase in gauging point spacing was lower than 1 mm. 

### 2.2. Neutron-Activation Analysis

The specimens for activation analysis were oven-dried at 105 °C for 24 h and they were ground into a 75 µm-sieve powder. Manual crushing in glazed mortar has been used to avoid any redundant impurities, which could influence the final neutron activation analysis (NAA) results. 

The determination of the concentrations of the elements and assessment of residual radioisotopes using neutron activation analysis was performed. Neutron activation analysis using the detection of delayed gamma rays originating from the (*n*, γ)-reaction of the irradiated nuclides, for the quantitative composition analysis of unknown samples. The selective measurement of the radiation of isotopes with different half-lives gives quantitative and qualitative information about the produced radioactive atoms. The NAA is especially capable of residual radioisotope determination in the µg/g concentration range, or below for 30–50 elements, depending on the nuclear properties of the elements of interest, the measurement conditions, the neutron flux, density and, in some cases, the matrix composition. For the elemental analysis, the k0-standardization method was used [26], which does not require a standard for the analysis. For NAA measurement, approximately 100–150 mg powdered samples of each were ampouled in high-purity quartz (Suprasil AN, Heraeus). The quartz ampules were wrapped in aluminum foil and encapsulated in an aluminum container. The 3 h irradiation was performed in a rotating, well-thermalized channel of the Budapest Research Reactor [27]. Together with the samples, flux-monitor foils of Au, Zr, Fe or Ni were co-irradiated, which are essential for the concentration calculations by the k0-method [28]. The thermal equivalent neutron flux in the rotating irradiation channel (No. 17) was 1.86 × 1013 cm^2^ s^−1^. The gamma rays emitted from the samples were counted with a high-purity germanium detector (ORTEC PopTop 55195-P HPGe) within iron low-level counting chambers to reduce the room background. The detector was connected to a dual-input ORTEC DSPEC 502 spectrometer and read out by the ORTEC Maestro 7 software. The spectra with 2 × 16 k channels were recorded with the zero-dead time (ZDT) option to accurately account for the different time-dynamics of the isotopes of interest. For spectrum evaluation, HyperLab 2013.1 software was used [29]. For the identification of radioactive isotopes and for element concentration calculations, the KayZero for Windows 3.06 program [30] was applied.

### 2.3. Expansion Testing

Recently, in 2018, methods for testing the potential alkali-silica reactivity of aggregates developed consistently with the relevant ASTM standards and RILEM recommendations, were adopted in Poland. The methods were: (a) accelerated method of investigating the expansion of mortar bar specimens in a 1 M NaOH solution at a temperature of 80 °C, and (b) a method of investigating the expansion of concrete prism specimens in a highly humid environment (RH > 95%) at a temperature of 38 °C [31]. 

In all of the test Portland cement, CEM I with the possibly highest alkali content available on Polish market was selected to satisfy the requirement Na_2_O_eq_ = 0.9 ± 0.1% [31].

Three mortar bar specimens 25 × 25 × 285 mm^3^ were prepared for each aggregate, which were processed by crushing and sieving to the appropriate gradation. An aggregate-to-cement ratio of 2.15 and water-to-cement ratio (by weight) of 0.47 were maintained. After 24 h in the mould, the mortar bars were stored for the next 24 h in water in 80 ± 1 °C. After that, their initial zero readings were recorded by a digital extensometer before immersion in 1 M NaOH at 80 ± 1 °C. Subsequent measurements were recorded at least 3 times up to 14 days.

For the long-term expansion test, all concrete prisms were prepared according to the guidelines [31] adapted from ASTM 1293 [32] and RILEM AAR-3 [33]. A cement with an initial alkali content of 0.88% was used. The water-to-cement ratio was 0.45. The content of cement was 420 kg/m^3^ with the equivalent of Na_2_O adjusted to 1.25% of the cement mass, which resulted in 5.25 kg of alkali per 1 m^3^ of concrete. The fine test aggregate was combined with a non-reactive coarse aggregate. The amphibolite was used as a non-reactive coarse aggregate, the mortar bar expansion after 14 days was 0.04%. The aggregate proportions were 30% of fine aggregate (0÷4 mm) and 70% of coarse aggregate (4÷22.4 mm). After casting prismatic 75 × 75 × 285 mm^3^ specimens, the specimens were protected from moisture loss and stored in the molds at 20 ± 2 °C for 1 day. Then, the prisms were demolded and stored in high humidity conditions at 38 ± 1 °C. The expansion result is an average of three specimens measured at 7, 28, 56, 90, 180, 270, and 365 days. An expansion limit of 0.040% at the end of the 1-year test was specified. 

The level of risk of alkali-silica reaction depends upon the importance of the concrete structure and the anticipated exposure conditions. In the concrete designed for radiation shields, ASR cannot be tolerated. Given the low-risk tolerance for NPP structures, the class SC4 according to ASTM C1778 [34] as taken into consideration.

### 2.4. Scanning Electron Microscopy and Energy-Dispersive Spectrometry (SEM-EDS)

A thorough investigation, using a combination of Scanning Electron Microscopy (SEM) in backscattered mode and Energy Dispersive Spectrometry (EDS), was conducted on pieces of mortar or concrete removed from selected specimens subjected to the conditions of accelerated or long term testing methods. The microstructural analysis was performed on the 25 × 42 × 10 mm^3^ specimens cut from the mortar bars or concrete prisms. The specimens were retrieved by slicing the bars/prisms using a slow speed diamond saw cooled by mineral oil. The specimens were then dried in an oven at 50 °C for 3 days and vacuum-impregnated with a low-viscosity epoxy.

Then, the specimens were placed in an oven maintained at 60 ± 2 °C to allow for polymerizing and the hardening of the medium overnight. After that time, a maximum of one millimeter was cut off the top of the specimen to expose a smooth, fresh surface on the face of interest. The cutting was performed using a Buehler slow-speed saw using mineral oil as the working fluid. The exposed specimen surface was lapped using lapping wheels with diamonds of specific sizes impregnated into the soft metal. Then, three wheels of increasingly fine diamond size were used: 45, 30, and 15 µm, respectively. Finally, diamond paste of the fineness 9, 6, 3, 1, and 0.25 µm sizes was spread on the fabric and diluted with an extender to reduce the viscosity of the paste. The each of the specimen surfaces was carefully observed under an optical microscope before proceeding to the next step of polishing. A strip of conductive tape was then attached to each polished sample, after which they were coated with a thin layer (15 ± 5 nm) of carbon for about a minute using the Quorum Q150R sputter coater. Each of the specimens was thoroughly examined using JEOL JSM-6380 LA SEM-EDX in the backscatter mode using an acceleration voltage of 15 kV. The instrument was equipped with a X-Max detector type SDD with 150 mm^2^ of active area. The instrument was used uncalibrated. Elements were identified by their respective Mn Kα-lines. Between examinations, the specimens were stored in a vacuum desiccator to protect them from laboratory humidity.

## 3. Results 

### 3.1. Neutron-Activation Analysis

In all tested aggregates and cements, the long-lived residual radioisotopes ^60^Co and ^152^Eu were found. Radioactive analysis revealed that ^60^Co and ^152^Eu were the major dominant long-lived residual radioisotopes and they have been chosen for further analysis. The results of the ^60^Co and ^152^Eu isotope contents in analyzed aggregate are presented in Figure 1 and Figure 2. The analysis of ^60^Co and ^152^Eu content in cements is presented in Figure 3. 

Among the coarse aggregate, melaphyre M1 (an igneous rock) and greywacke GW1 (a sedimentary rock) aggregates were characterized by the highest content of ^60^Co and ^152^Eu, more than 15 ppm and 1.5 ppm, accordingly. The lowest values for ^60^Co and ^152^Eu concentrations among all coarse aggregates has been found for the limestone L1 aggregate (a sedimentary rock)—0.20 ppm and 0.05 ppm. Comparatively low contents of these isotopes were found in baryte B1 aggregate, 0.77 ppm for ^60^Co and 0.38 ppm for ^152^Eu. Fine aggregate contained significantly less ^60^Co and ^152^Eu compared to coarse aggregate—more than ten times less. The lowest content of ^60^Co and ^152^Eu in fine aggregate was found in limestone sand (S1), respectively, 0.44 ppm and 0.13 ppm. The difference resulting from the origin (composition) of quartz sand is clearly visible. Higher ^60^Co and ^152^Eu concentration values were obtained in tests with fossil sand (S4–S7) compared to river sand (S2 and S3)—less than 1.5 ppm ^60^Co concentration and less than 0.20 ppm ^152^Eu concentration in river sands in comparison for 2.0 ppm and 0.35 ppm for fossil sands. 

The results from ^60^Co and ^152^Eu analysis content in cements clearly indicate the difference between ordinary Portland cements C1, C2 and C3 (Fe_2_O_3_ = 3.11 ± 0.15%) and cement (C4) with reduced iron content (Fe_2_O_3_ = 0.30%). The content of ^60^Co in Portland cements was between 7.9 and 11.3 ppm, while in cement C4, this was only 1.8 ppm, almost ten times less. The content of ^152^Eu also significantly differed; in cements C1–C3, the content was 0.59 ± 0.03 ppm, and in cement C4, this was 0.25 ppm. The ^60^Co content in Portland cements (C1–C3) was comparable to the content of this isotope in quartzite coarse aggregate (Q1), 9.02 ppm for ^60^Co. 

### 3.2. Expansion Results

The results of the accelerated mortar bar expansion after 14 days of testing in 80 °C and 1 M NaOH are presented in Figure 4 and Figure 5. The coarse aggregates, which were characterized by expansion of mortar bars of less than 0.1%, were further analyzed according to long-term prism testing at 38 °C, and high humidity conditions. The results of concrete prism expansion after 365 days are presented in Figure 6. 

The results of the accelerated mortar bar test showed that the lowest expansion was achieved by mortars bar made with barite B1 and limestone L1 aggregate, after 14 days 0.01 and 0.03%, respectively. Granite aggregates (G1 and G2) also did not reveal alkali-silica reaction potential, G1—0.05% and G2—0.08% after 14 days. The mortar bars with quartzites Q1 and Q2, greywacke GW1 and flint F1 aggregates showed very fast expansion; more than 0.1% after 5 days of testing. Quartzite (Q1 and Q2) turned out to be a very reactive coarse aggregate, expansion over 0.36%. All other aggregates: melaphyre M1, greywacke GW1, flint F1 showed susceptibility to alkali-silica reaction (respectively, 0.19, 0.31 and 0.19%). 

The susceptibility of river sands to ASR is clearly visible among fine aggregate, Figure 5 The highest expansion was achieved by mortars made with river sands S2 and S3 (0.32% and 0.24%), much more above the allowable limit, 0.1%. The lowest value of expansion was for mortar bar with limestone sand S1, at an expansion of 0.01%. Mortars with fossil sands S6 and S5 were also characterized by expansion below 0.1%, respectively 0.04 and 0.08%. 

It was assumed that, if the expansion of the mortar bars after 14 days was much more than 0.3% (quartzite Q1 and Q2, greywacke GW1) or 0.2% (melaphyre M1, flint F1), the aggregate was highly susceptible to an alkali-silica reaction. The expansion of granite G1 and G2 and limestone L1, as well as baryte B1 mortar bars, was less than 0.1% after 14 days, so these aggregates were further tested according to long-term prism test at 38 °C and high humidity conditions. 

The expansion curves for all concretes after a year did not exceed the limit of 0.040%. The highest expansion was characterized by concrete with granite aggregate (G1—0.035%, G2—0.028%), then limestone aggregate (L1—0.027%) and the lowest expansion: concrete with baryte (B1—0.026%). The different shape of the curves for each of the examined aggregates is visible. The fastest expansion was shown by concrete with granite G1, unlike concrete with granite G2, which showed the latest beginning of expansion. Up to 90 days, concrete with baryte B1 showed a rapid increase in expansion, but thereafter significant extinction is noticeable. 

The result of the long-term prism test confirmed observations according to accelerated mortar bar test. The aggregate granite G1 and G2, limestone L1 and baryte B1 were classified as non-reactive (expansion below 0.04% after one year).

### 3.3. SEM Microstructure Analysis

After the accelerated expansion test mortar bars were selected and the microstructure analysis was performed on polished specimens. The observed damage due to ASR was various depending on the type of aggregate. The evidence of alkali-silica reaction—cracking and microcracking occurred in the specimens containing greywacke GW1, melaphyre M1, flint F1 and quartzite aggregate Q1 and Q2, as well as siliceous sand S2, S3, S4 and S7. The results of the mortar microstructure SEM analysis are consistent with the results of the preceding expansion measurements. In Figure 7 and Figure 8, the SEM microphotographs documenting the presence of an alkali-silica gel in greywacke GW1 and siliceous sand S3 and, thus, the reason for mortar bar expansion, are presented. The Si-Ca-Na-K gel, which partially filled the air pore, is the result of ASR in greywacke aggregate GW1, Figure 7. The alkali-silica gel passed through whole grains of siliceous sand S3 and then into the cracks and air-voids, as shown in Figure 8. 

From all analyzed specimens after long-term prism test, only granite G1 showed slight signs of an alkali-silica reaction; Figure 9. At the boundary of orthoclase and quartz, the micro-area analysis showed the presence of trace amounts of Si-Ca-Na-K gel in the cracked granite G1 aggregate. The cracking was not so great that the expansion of the concrete prisms after one year of exposure at the temperature of 38 °C exceeded 0.04%. However, granite aggregate G1 tends to show a slow alkali-silica reaction potential.

## 4. Discussion

The dominant long-lived residual radioisotopes induced in concrete, which occupy 99–100% of the total residual radioactivity, are ^60^Co, ^152^Eu and ^154^Eu [4]. Current investigations, based on NAA, revealed that the ^60^Co and ^152^Eu isotope contents in analyzed aggregate partly coincide with results for Japanese [5,35] and Indian [36] research; Figure 10. Various rocks were collected from all over Japan and analyzed by Suzuki et al. [5]. They analyzed the trace elements of ^60^Co, ^152^Eu and ^134^Cs. Kinno et al. [35] determined the Co and Eu contents in the several raw materials for concrete, as well as Pai et al. [36]. They have analyzed eleven types of coarse aggregates from different geological formations. From the results of screening tests for neutron irradiation, Kinno et al. [35] found that the aggregate for low-activation concrete was quartzite with the lowest concentration of ^60^Co = 0.1 ppm and ^152^Eu = 0.004 ppm and limestone (^60^Co = 0.08 ppm and ^152^Eu = 0.05 ppm). According to Suzuki et al. [5], the aggregate made from quartzite rock exhibited the lowest induced activity on neutron irradiation among all tested silica aggregates. The mean concentration of ^60^Co was 3.55 ppm and ^152^Eu 0.297 ppm. Pai et al. [36] made similar observations in their research. They achieved the lowest concentration of ^60^Co and ^1^^52^Eu, in quartzite 0.79 ppm and 0.12 ppm, respectively. Kimura et al. [3] presented the distribution of quantities for Europium and Cobalt in aggregates and they showed that only fused alumina and quartzite aggregate were within the curve $\overset{3}{\underset{}{\sum }}c_{i}/c_{li}$ = 0.1 for the activation. Silica sand, baryte and limestone aggregate were within the curve equal 1.0. 

The current investigation revealed that the concentrations of the ^60^Co and ^152^Eu isotopes in coarse aggregate from igneous rocks were also relatively high, but they significantly differ from one other. In rock aggregate of plutonic origin (granite) the average concentration of ^60^Co was 7.5 times lower than in the rock aggregate of volcanic origin (melaphyre), 3 ppm and 22.6 ppm, respectively. Similar results were obtained for ^152^Eu content, 0.6 ppm for granite and 2.6 ppm for melaphyre. The lowest values of ^60^Co and ^152^Eu in all coarse aggregate were achieved for limestone and baryte aggregates. On the other hand, greywacke and flint revealed the highest content of above residual radioisotopes. All these aggregates belong to the group of sedimentary rocks. A more accurate separation of sedimentary rocks into clastic and organogenetic showed that greywacke, which belongs to the clastic group cannot be taken into consideration as an aggregate for low-activation shielding concrete. Pai et al. [36] showed that, from eleven analyzed coarse aggregates, the quartzite had the lowest potential of generating long-lived gamma-emitting radionuclides. The content of ^60^Co and ^152^Eu in quartzite was accordingly eight and three times lower than in limestone aggregate. The current investigation on coarse aggregate revealed the average content of ^60^Co and ^152^Eu in quartzite of 5.7 and 1.0 ppm, respectively, which was higher than the content of such isotopes in limestone and baryte as well as in granite aggregate.

Smaller differences in the content of ^60^Co and ^152^Eu were obtained for fine aggregate than in coarse aggregate, which resulted from larger diversity of rock origin of coarse aggregate, Figure 10. 

All above results indicated that quartzite could be an appropriate aggregate to be used as constituent material for low-activation concrete for neutron shielding elements. However, in recent research, performed by Šachlová et al. [37] and Jensen and Sujjavanich [38], attention has been drawn to quartzite, due to varying alkali-silica reaction enhancement properties. There are varying factors that affect the ASR susceptibility of quartzite, including the degree of deformation and the size of the grain quartz [37] and close spacing of quartzite, feldspar and muscovite grains. Jensen et al. [38] reported the occurrence of the alkali-silica reaction in concrete containing quartzite as coarse aggregate. The individual quartzite crystals were in the size range of 10–50 μm. The above-described aggregate was sourced from Thailand, yet the author notes that the products of alkali-silica reaction and extensive microcracks were characteristic of slowly reacting aggregates, such as the quartzite he had observed in Norway. 

The results of the accelerated mortar bar test showed that the highest expansion was achieved by mortar bars made with quartzite (Q1 and Q2) aggregate. They turned out to be a very reactive aggregate, with expansion of over 0.36%. Baryte B1 and limestone L1 aggregates did not show susceptibility to provoking ASR, while all other aggregates— melaphyre M1, greywacke GW1, flint F1—showed susceptibility to alkali-silica reaction. The expansion of the concrete prisms made with baryte B1, limestone L1 and granite G1 and G2 aggregate after one year of exposure in 38 °C was lower than 0.040%. However, granite aggregates G1 showed a trace amount of alkali-silica gel in cracked aggregate grains observed after one year of exposure at 38 °C.

The susceptibility of river sands to ASR is clearly visible among fine aggregate. The highest expansion was achieved by mortars made with sand S2 and S3 (0.32% and 0.24%), much higher than the allowable limit of 0.1%. Previous investigation revealed that certain hematite aggregate or impure baryte aggregate may also be highly reactive [39,40]. 

Blended cements, especially fly ash cement, are known to reduce the potential of alkali-silica reaction in concrete [41,42], but they have not been taken into consideration in this investigation due to their activation ability and the high natural radionuclides concentration, [5]. The chemical composition of analyzed Portland cements (C1, C2 and C3) was similar, excluding the alkali content. It varied from 0.56 to 1.12% of Na_2_O_eq_. The conducted research has shown that ^152^Eu content in Portland cements was similar, but the ^60^Co concentration significantly depending on the Na_2_O_eq_ content; Figure 11a. The lower the alkali content in cement, the lower the ^60^Co concentration. Additionally, it has been noted that the content of iron oxide in cement clearly influenced the concentration of ^152^Eu, Figure 11b. The concentration of ^152^Eu increased with increasing Fe_2_O_3_ content in cement.

As shown in [5], the concentration of ^152^Eu in plutonic rock and volcanic rock as aggregates can be higher than of ^60^Co. It was also suggested that the use of sedimentary rock as aggregates would reduce the content of ^152^Eu in biological shielding concrete. 

At the decommissioning stage of NPP, the activation level of elements in concrete should be as low as possible to ease necessary waste disposal procedures. Therefore, concrete constituents should contain, as little as possible, the residual radioisotopes with large activation cross section. On the other hand, concrete structures in NPPs and nuclear facilities are directly classified as structures that require the highest level of ASR prevention and no risk due to ASR being currently tolerated (ASTM C1778 [34]). The aggregate and cement should be carefully selected to be inert to the alkali-silica reaction while providing the smallest possible concentration of major dominant long-lived residual radioisotopes. However, it is suggested in recent literature that neutron radiation could significantly increase the alkali reactivity of silica-rich aggregates [43]. The reason for neutron-enhanced ASR in concrete could be associated both with radiation-induced volume expansion of minerals as well as enhanced dissolution of silica, as suggested in [44,45]. If it is confirmed for a variety of rock aggregates an optimal concrete mix composition, established considering the proposed criteria of the designed durability while maintaining the minimum content of residual radioisotopes, ^60^Co and ^152^Eu would not be unambiguous. The radiation-induced deterioration of concrete is still not well understood; it is thought to take place mostly at high neutron fluence levels, possibly beyond the threshold level of 1019 *n*/cm^2^ [46]. With the future development of knowledge in this area, the mix design for durability should be supplemented with the criterion of susceptibility of concrete constituents to radiation-induced damage.

## 5. Conclusions

The following conclusions can be drawn:

The concentration of ^60^Co and ^152^Eu activated by neutron radiation in natural fine aggregate was lower than in natural coarse aggregate. The average content of ^60^Co and ^152^Eu in natural siliceous fine aggregate amounted to 1.71 ± 0.38 ppm and 0.30 ± 0.09 ppm, respectively. It was 8.73 ± 7.98 ppm and 1.05 ± 0.83 ppm in natural coarse aggregate.The influence of the sand origin on the ^60^Co and ^152^Eu content was clearly visible in the analyzed natural fine aggregate. Lower concentrations of Cobalt and Europium were present in the river sands, compared to fossil fine aggregate.The lowest values of ^60^Co and ^152^Eu concentration were found in limestone, both in fine aggregate, at 0.44 ppm and 0.13 ppm, and in coarse aggregate, at 0.20 ppm and 0.05 ppm, respectively.For the considered range of Portland cements CEM I 42.5 R and 52.5 R the content of ^60^Co was proportional to the content of Fe_2_O_3_ and the content of ^152^Eu was proportional to the total content of alkalis.Quartzite and greywacke aggregates were found to be highly reactive in the alkaline environment of Portland cement concrete.Due to the high potential of alkali-silica reaction, it is not recommended to use quartzite as a coarse aggregate, as well as siliceous river sand as a fine aggregate for shielding concrete, despite the low contents of ^60^Co and ^152^Eu. At the same time, it is suggested to use the cement with the lowest alkali content, both due to the possibility of alkali-silica reaction of the aggregate in concrete, as well as a lower content of ^60^Co.Limestone with low content of siliceous minerals is good for preventing the alkalisilica reaction and the formation of ^60^Co and ^152^Eu radioisotopes.

## Figures and Tables

**Figure 1 materials-14-04595-f001:**
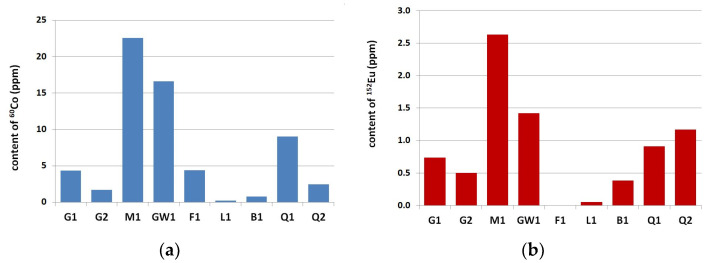
Content of (**a**) ^60^Co and (**b**) ^152^Eu in coarse aggregate.

**Figure 2 materials-14-04595-f002:**
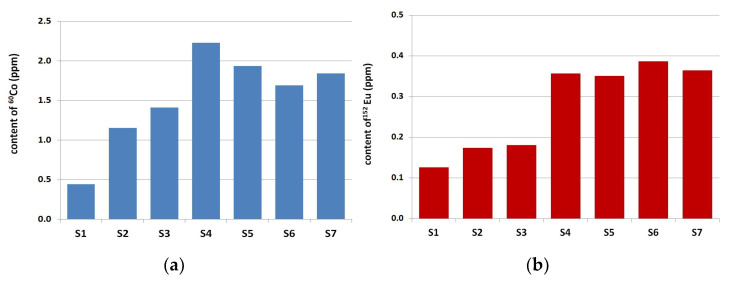
Content of (**a**) ^60^Co and (**b**) ^152^Eu in fine aggregate.

**Figure 3 materials-14-04595-f003:**
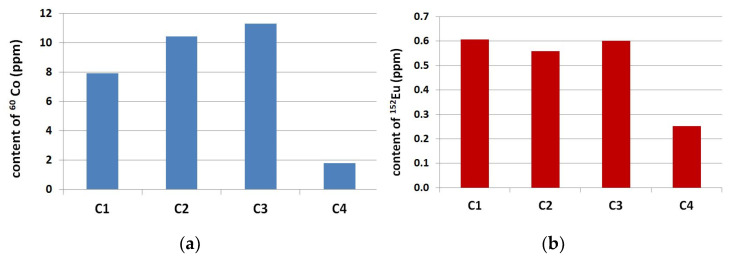
Content of (**a**) ^60^Co and (**b**) ^152^Eu in Portland cements.

**Figure 4 materials-14-04595-f004:**
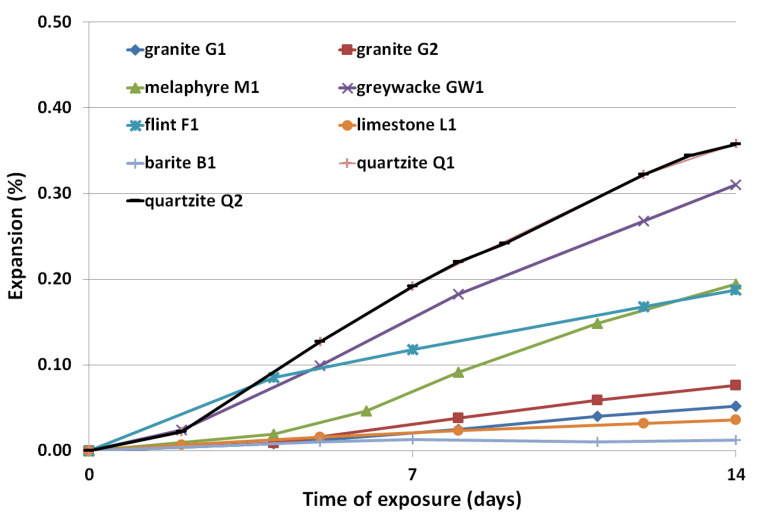
Expansion of mortar specimens with crushed coarse aggregates as a function of exposure time in 1 M NaOH solution at the temperature of 80 °C.

**Figure 5 materials-14-04595-f005:**
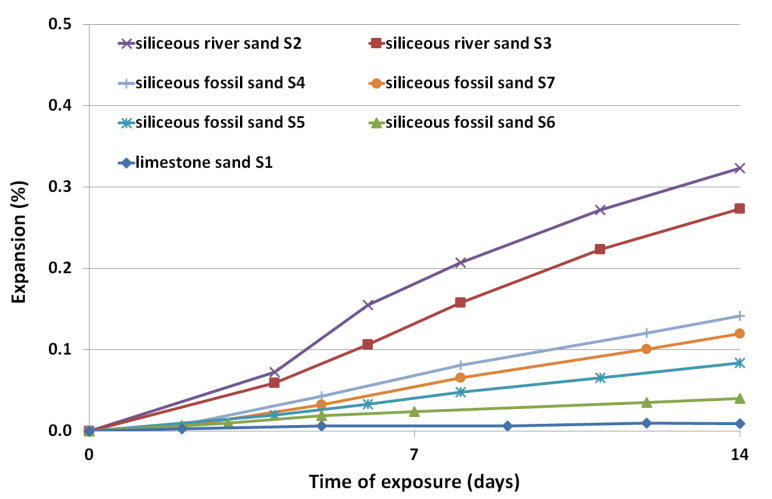
Expansion of mortar specimens with fine aggregate as a function of exposure time in 1 M NaOH solution at the temperature of 80 °C.

**Figure 6 materials-14-04595-f006:**
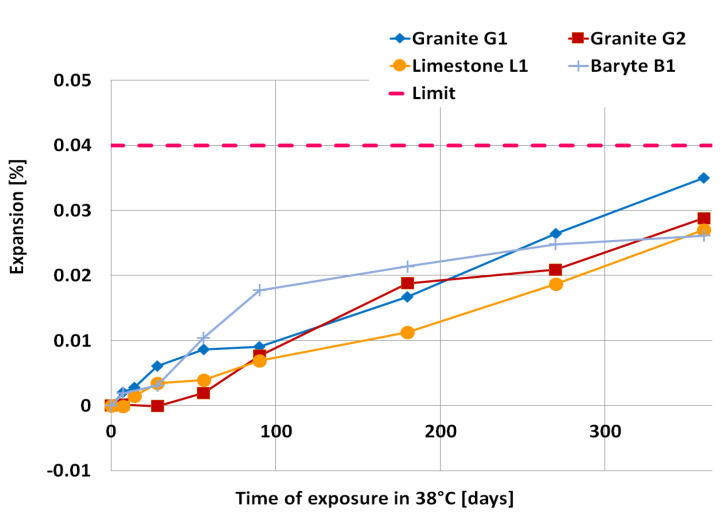
Expansion of concrete prism specimens as a function of exposure time at the temperature of 38 °C.

**Figure 7 materials-14-04595-f007:**
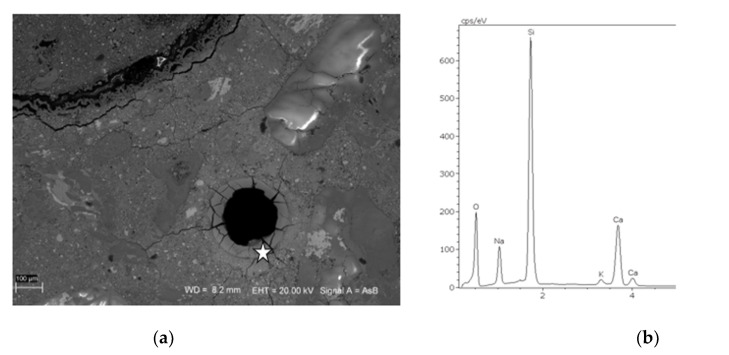
Microphotograph of a cracked matrix and greywacke grains (**a**) with EDS analysis of the Si-Ca-Na-K gel lining air-void (**b**); mortar GW1, SEM analysis after accelerated mortar bar test.

**Figure 8 materials-14-04595-f008:**
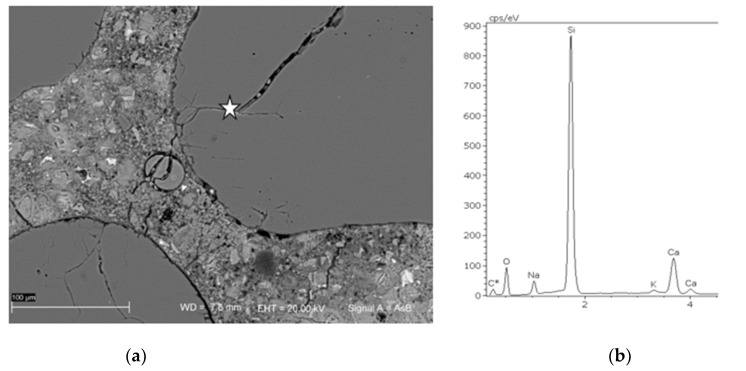
Microphotograph of a cracked siliceous sand (**a**) with EDS analysis of the Si-Ca-Na-K gel coming out of the sand grains and completely filling the air-void (**b**); mortar with sand S3, SEM analysis after accelerated mortar bar test.

**Figure 9 materials-14-04595-f009:**
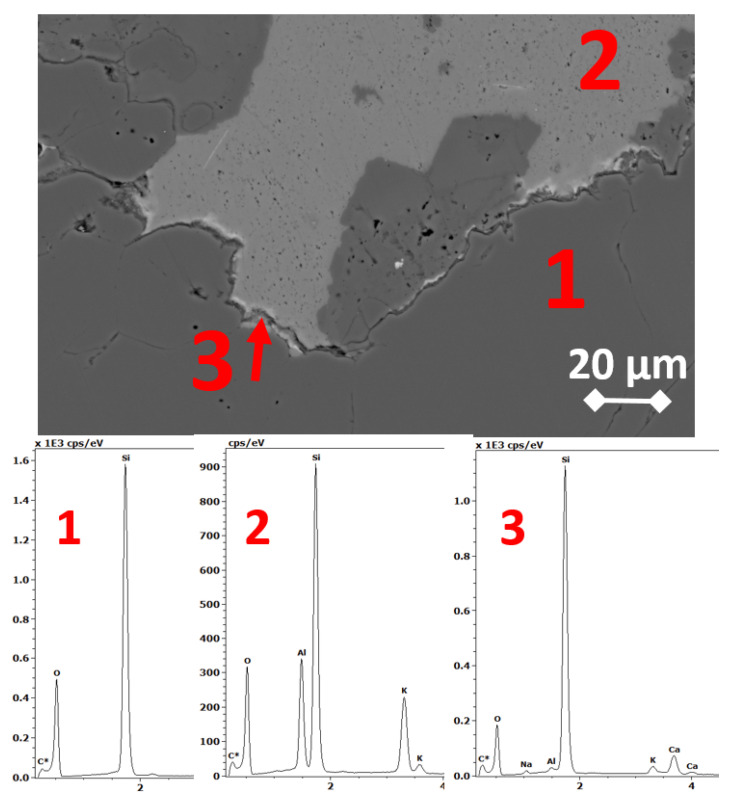
Microphotograph of a cracked granite G1 aggregate with EDS analysis of the trace of Si-Ca-Na-K gel: (**1**) quartz, (**2**) orthoclase, (**3**) traces of Si-Ca-Na-K gel (SEM analysis after long-term concrete prism test).

**Figure 10 materials-14-04595-f010:**
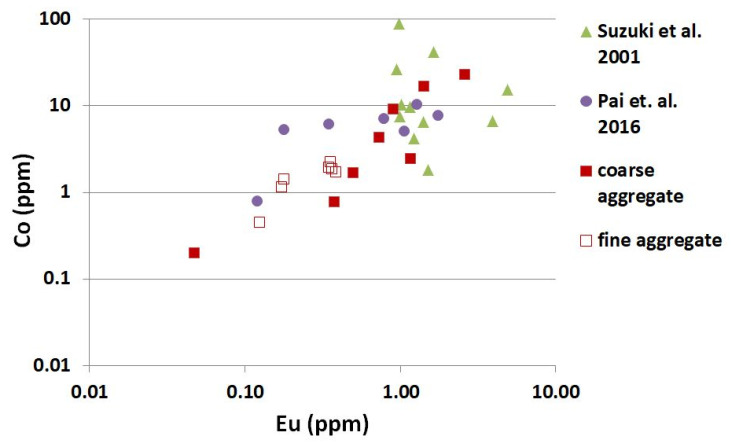
Content of Cobalt and Europium in mineral aggregate determined in the current and previous research.

**Figure 11 materials-14-04595-f011:**
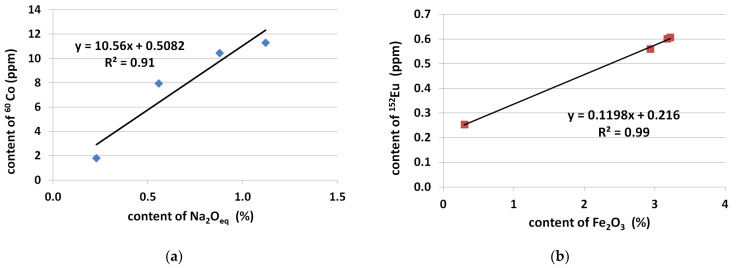
Concentration of the long-lived isotopes depending on: (**a**) the alkali content, and (**b**) Fe_2_O_3_ content in cement.

**Table 1 materials-14-04595-t001:** Designation and density of analyzed crushed coarse aggregate.

Aggregate	Designation	Density, g/cm^3^ [21]
quartzite	Q1	2.62
	Q2	2.60
granite	G1	2.63
	G2	2.64
flint	F1	2.65
melaphyre	M1	2.70
greywacke	GW1	2.70
limestone	L1	2.71
baryte	B1	4.20

**Table 2 materials-14-04595-t002:** Chemical composition of cements, data from the cement plants, XRF, wt. %.

Constituent	C1	C2	C3	C4
	CEM I 42.5R	CEM I 52.5R	CEM I 42.5R	CEM I 52.5R
SiO_2_	19.03	19.42	19.43	24.40
Al_2_O_3_	4.84	5.45	4.84	2.11
Fe_2_O_3_	3.22	2.94	3.18	0.30
CaO	63.64	64.1	61.81	68.40
MgO	1.15	1.75	2.56	0.66
SO_3_	2.97	3.5	3.93	2.09
Na_2_O	0.21	0.24	0.41	0.17
K_2_O	0.53	0.97	1.08	0.09
LOI	3.34	2.50	2.67	1.22

## Data Availability

The data presented in this study cannot be shared at this time as the data also forms part of an ongoing study.

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
