# Peer review of "Assessment of Long Lived Isotopes in Alkali-Silica Resistant Concrete Designed for Nuclear Installations"

_materials, 2021, doi:10.3390/ma14164595_

Round 1

Reviewer 1 Report

[Materials] Manuscript ID: materials-1293318

Materials

Assessment of Long-Lived Isotopes in Alkali-Silica Resistant Concrete Designed for Nuclear Installations

Reviewer comments:

SUMMARY

The manuscript deals with an investigation on the radiological characterisation of Alkali-Silica Resistant Concrete. This is a topic that has not been widely covered in the literature, therefore, this a subject of great interest, but it is somehow limited in the analysis and application of these results.

MAIN IMPRESSIONS

This paper has an undeniable practical usefulness. However, from a scientific point of view, the following issues must be addressed: i) The current state of the research field should be carefully reviewed, and key publications cited, ii) the conclusion should be improved.

MORE DETAILED COMMENTS

Line 5: ”1, 3, 4, 5 Institute of … “. Numbers from 1-5, except “2”, may replaced by “1” for all the cases, except “2”.

Line 66: Nested References [8][9][10][11]. Could you please to explain the content of each reference?

Line 71: Why was it not possible to get across better the advantages of supplementary cementitious materials on ASR mitigation? I suggest discussing the following reference: Durability of Blended Cements Made with Reactive Aggregates. Materials 2021, 14, 2948. https://doi.org/10.3390/ma14112948

The use of some Portland cement constituents is effective in mitigating ASR deleterious effects. The reasons for mentioned positive performance can be found in: Sustainable and Durable Performance of Pozzolanic Additions to Prevent Alkali-Silica Reaction (ASR) Promoted by Aggregates with Different Reaction Rates. Appl. Sci. 2020, 10, 9042. https://doi.org/10.3390/app10249042

Line 76: Could you please add a reference? For instance, Alkali-silica reaction of aggregates for concrete pavements in Chihuahua’s State, Mexico. Materiales De Construcción, (2002) 52(268), 19–31. https://doi.org/10.3989/mc.2002.v52.i268.314

Line 88: As the aim of the paper is to define a “new criterion of the content of long-lived residual radioisotopes in concrete constituents”, you should introduce the radiological activity concentration of the concrete constituents, at least Portland cement and additions. For instance: Radiation dose calculation of fine and coarse coal fly ash used for building purposes. J Radioanal Nucl Chem 327, 1045–1054 (2021). https://doi.org/10.1007/s10967-020-07578-8 ;

 Line 113:  Could you please add the origin (factory, company, location) of the Four Portland cements?

Line 113:  Could you please complete the CEM I designation? For instance, CEM I 42.5 R.

Line 117: loss on ignition could be included in Table 2 with reference to EN 196-2 [23].

Line 124: Could you please add the referemce for Le Chatelier method?

Line 126: “The specimens for activation analysis were selected from all tested aggregates”, but in Line 218 you say “In all tested aggregates and cements the …”, and in Line 248:  Figure 4. Content of 60Co and 152Eu in Portland cements

216-320: 3.1. Neutron activation analysis: Could you please compare these results with the ones found in the literature?

Line 332: According to Kinno et al. [33] the aggregate for low-activation concrete was quartzite with the lowest concentration of 60Co and 152Eu, and according to Suzuki et al. [5] the aggregate made from quartzite rock exhibited the lowest induced activity on neutron irradiation among all tested silica aggregates. Could you please compare these findings with your results and discuss them?

Line 345: Could you please add an explanation about why ”In rock aggregate of plutonic origin (granite) the average  e concentration of 60Co was 7.5 times lower than in rock aggregate of volcanic origin”?

Line 365: Probably, after the sentence “All above results indicated that quartzite could be an appropriate aggregate to be used as constituent material for low-activation concrete for neutron shielding elements.”, could be convenient to add “However, …”, i.e.:

“ … for low-activation concrete for neutron shielding elements. However, in recent research performed by Šachlová et al. [35], and Jensen and Sujjavanich [36] …”.

Lines 428-452: Conclusions: The main original findings should be highlighted.

Line 447: This conclusion is not supported by the experimental results. “Due to the high potential of alkali-silica reaction, it is not recommended to use quartzite as coarse aggregate as well as siliceous river sand as fine aggregate.” You should test ASR and discuss your results. Quartzite is used in concrete. Could you please check the following papers?:

  • Tufail, M., Shahzada, K., Gencturk, B. et al. Effect of Elevated Temperature on Mechanical Properties of Limestone, Quartzite and Granite Concrete. Int J Concr Struct Mater 11, 17–28 (2017). https://doi.org/10.1007/s40069-016-0175-2
  • Kavitha, K. & Cheela, Venkata & SSSV, Gopala. (2019). Utilisation of Quartzite as Fine Aggregate in Concrete. https://doi.org/10.18780/e-jst.v10i5.3084
  • Matéria (Rio J.) 24 (4) • 2019 • https://doi.org/10.1590/S1517-707620190004.0855

Line 449: Why ”... it is suggested to use cement with the lowest alkali content, both due to the possibility of alkali-silica reaction of the aggregate in concrete as well as a lower content of 60Co… ”? Firstly, all the Portland cements tested worldwide are safe from a radiological point of view. Secondly, you do not need cement with the lowest alkali content when good quality aggregates are used. Accordingly, you are recommending the use of cement with the lowest alkali content, which has a higher production cost.

Line 451: This conclusion is not supported by the experimental results. “Limestone with low content of siliceous minerals is good for preventing the alkali-silica reaction …”, but what about Alkali-carbonate reaction (ACR)? The deterioration caused by alkali-carbonate reactions is similar to that caused by ASR, however, ACR is relatively rare.

Line 499: In ref. [22] PN-EN 197-1:2012, I suggest using the following format because it is an European standard:

  • European Committee for Standardization (CEN). EN 197−1:2011. Cement—Part 1: Composition, Specifications and Conformity Criteria for Common Cement; European Committee for Standardization (CEN): Brussels, Belgium, 2011.

Lines 501-504: The same comment for references 23 & 24.

Line 556: Competing interests must go before the references.

Line 558: Funding must go before the references.

Line 562: Authors' contributions must go before the references.

RECOMMENDATION

In conclusion, Minor changes have been proposed.

Author Response

Dear Reviewer,

We highly appreciate your comments concerning our manuscript. We have studied your comments carefully and have introduced appropriate corrections, which we hope will be met with your approval. Please, find below our detailed answers to your comments and suggestions.

Sincerely,

Daria Jóźwiak-Niedźwiedzka

Reviewer 2 Report

The topics addressed in this manuscript are interesting. The authors should consider following items to improve the quality of the manuscript:

  1. Not common to have “,” before citation.
  2. I’m not sure the Fig. 1 provides added values, given that from ref. 4.
  3. Lines 72-73: not clear what you mean, revise please
  4. Lines 81-82: Why do you refer to modelling. If it is a topic to deal with in this paper, you should extend the discussion.
  5. Materials section: not clear what density you reported. And why do you add uncertainty to S4-S7 sands, but not for others?
  6. What is the basis to propose the concrete mix design in this study? You mentioned in the title that concrete designed for nuclear installation, but I don’t see a clear description in the body text.
  7. Lines 123-124: don’t understand, please revise
  8. What I’m missing in this paper is that how neutron irradiation would affect the ASR. You did not perform ASR tests on irradiated concretes/mortars.
  9. The size of coarse aggregates is not consistent between lines 179 and 104.
  10. Line 198: is there any specific reason to cure the impregnated samples at 60oC? This would induce microcracks.
  11. Lines 279-281: not clear, where should I see the results for ASR test on mortar samples at 38oC?
  12. Lines 396-402: nice to have these relationship, but what is the basis to propose these relations?
  13. Lines 416-418: This is what I’m looking for in this work, but your experimental plan was not designed for this.  

Author Response

(The authors gave the same response as above.)

Round 2

Reviewer 2 Report

The authors have addressed all the comments. The quality of this manuscript is improved, ready for publication.